# Use of Heteroatom-Doped g-C$_3$N$_4$ Particles as Catalysts for Dehydrogenation of Sodium Borohydride in Methanol

**Sahin Demirci [1] and Nurettin Sahiner [1,2,3,*]**

1    Department of Chemistry & Nanoscience and Technology Research and Application Center (NANORAC),
     Faculty of Science, Canakkale Onsekiz Mart University, Terzioglu Campus, Canakkale 17100, Turkey
2    Department of Chemical and Biomolecular Engineering, University of South Florida, Tampa, FL 33620, USA
3    Department of Ophthalmology, Morsani College of Medicine, University of South Florida,
     12901 Bruce B. Downs Blvd., MDC21, Tampa, FL 33612, USA
*    Correspondence: sahiner71@gmail.com or nsahiner@usf.edu

**Abstract:** Here, graphitic carbon nitride (g-C$_3$N$_4$) was synthesized from melamine, doped with heteroatoms, such as B, S, and P reported using boric acid, sulfur, and phosphorous red as dopants, respectively. The catalytic performances of g-C$_3$N$_4$, and heteroatom-doped g-C$_3$N$_4$ (H@g-C$_3$N$_4$ (H=B, S or P) particles as catalysts in the dehydrogenation of sodium borohydride (NaBH$_4$) in methanol to generate hydrogen (H$_2$) were investigated. The prepared g-C$_3$N$_4$-based structures were used as catalysts for hydrogen (H$_2$) production in the dehydrogenation reaction of sodium borohydride (NaBH$_4$) in methanol. The catalytic performance of H@g-C$_3$N$_4$ (H=B, S or P) structures in the dehydrogenation reaction of sodium borohydride (NaBH$_4$) in methanol was determined to be higher than the catalytic performance of the bare g-C$_3$N$_4$ structure. The hydrogen generation rate (HGR) values were calculated for the reactions catalyzed by B@g-C$_3$N$_4$, P@g-C$_3$N$_4$, and S@g-C$_3$N$_4$ as 609 $\pm$ 48, 699 $\pm$ 48, and 429 $\pm$ 55 mL H$_2$/g of cat.min, respectively, which is only 282 $\pm$ 11 mL H$_2$/g of cat.min for the native g-C$_3$N$_4$-catalyzed one. The activation energies (Ea) were found to be relatively low, such as 31.2, 26.9, and 31.2 kJ/mol, for the reactions catalyzed by B@g-C$_3$N$_4$, P@g-C$_3$N$_4$, and S@g-C$_3$N$_4$, respectively. In addition, in the reuse studies, it was concluded that B@g-C$_3$N$_4$, P@g-C$_3$N$_4$, and S@g-C$_3$N$_4$ catalysts can readily complete the reaction with 100% conversion, even in five consecutive uses, and afforded promising potential with more than 80% activity for each use.

**Keywords:** carbon-based catalyst; graphitic carbon nitride; g-C$_3$N$_4$; catalyst; H$_2$ production; NaBH$_4$ methanolysis

## 1. Introduction

One of the newest members of carbon-based structures that is two-dimensionally conjugated and a visible light-sensitive graphitic carbon nitride (g-C$_3$N$_4$) can be considered as a layered polymer formed via C- and N-atoms, and was scrutinized by many researchers due to its unique properties [1–4]. Because of its unique physical and chemical properties, including high surface area, excellent electrical conductivity, strong mechanical strength, unmatched thermal conductivity, ease of functionalization, tunable optics, etc., these nanomaterials are increasingly used in catalysis, energy storage, and the biomedical field [5–11]. Studies examining the catalyst properties of g-C$_3$N$_4$ structures are related to their photocatalytic properties, as they exhibit a band gap of ~2.7 eV [2,3,9,10]. The g-C$_3$N$_4$ structures are also reported to act as a highly efficient and photostable organic photocatalyst after doping with heteroatoms, such as S, B, O, and P [12–14]. Moreover, it was reported in the literature that g-C$_3$N$_4$ structures can be used as a catalyst for hydrogen (H$_2$) production from the dehydrogenation reactions of sodium borohydride (NaBH$_4$) in methanol after doping with heteroatoms [15,16]. In different studies, g-C$_3$N$_4$ structures prepared from the dicyandiamide molecule were doped with O by treating with HNO$_3$, and with P by treating with H$_3$PO$_4$ [15,16]. Subsequently, the potential of using g-C$_3$N$_4$

structures doped with O and P as a catalyst in the dehydrogenation reaction of $NaBH_4$ in methanol was investigated [15,16].

Since the $NaBH_4$ hydrolysis reaction has low conversion rates and reaction kinetics at low temperatures, researchers focused on finding new solvents for $H_2$ production from $NaBH_4$ using different solvents [17–19]. It was consequently seen that methanol can be a suitable solvent with a higher $H_2$ production rate and higher volume $NaBH_4$ solubility, even at low temperatures, which can be obtained from renewable resources or biomass raw materials [17,18,20–22]. In the presence of a suitable catalyst, the dehydrogenation reaction of 1 mole of $NaBH_4$ in methanol can produce 4 moles of $H_2$, as given in Equation (1):

$$NaBH_4 + 4CH_3OH \rightarrow 4H_2 + NaB(OCH_3)_4 \tag{1}$$

Various metal-containing or non-metallic catalysts are reported for the dehydrogenation reaction of $NaBH_4$ in methanol [23–26]. However, these catalysts are costly, as well as have some limiting factors, such as low recyclability, harmful effects on the environment, and low catalytic strength [27,28]. Therefore, a low-cost, highly stable, easily recyclable, and environmentally friendly alternative catalyst is required to produce $H_2$ from the dehydrogenation reaction of $NaBH_4$ in methanol.

In this study, $g\text{-}C_3N_4$ from melamine as a precursor doped with heteroatoms, such as B, S, and P, were reported using boric acid, sulfur, and phosphorous red as dopants, respectively. The catalytic performances of $g\text{-}C_3N_4$, and heteroatom-doped $g\text{-}C_3N_4$ (H@g-$C_3N_4$, H: B, S or P) structures as catalysts in the dehydrogenation of $NaBH_4$ in methanol to generate $H_2$ were investigated. The effects of heteroatom types and the reaction temperature on the catalytic activity of $g\text{-}C_3N_4$-based structures for $H_2$ production from $NaBH_4$ were investigated. The activation parameters, activation energy (Ea), enthalpy (ΔH), and entropy (ΔH) values were calculated for $g\text{-}C_3N_4$-based structure catalysts in $H_2$ generation reactions from $NaBH_4$. Moreover, the reusability of the $g\text{-}C_3N_4$-based catalyst in $H_2$ production reactions was investigated, and the catalytic performances of these materials were compared with the other catalysts used in the literature for the same purpose.

## 2. Materials and Methods

### 2.1. Materials

Melamine (99%, Sigma Aldrich, St. Louis, MO, USA) was used as a precursor to synthesize the graphitic carbon nitride ($g\text{-}C_3N_4$) structures. Boric acid (99.5%, Sigma Aldrich, St. Louis, MO, USA), phosphorous red (97%, Merck, Italy), and sulfur (Reagent grade, Sigma Aldrich, St. Louis, MO, USA) were used as B, P, and S sources for the doping of the $g\text{-}C_3N_4$ structures. Sodium borohydride ($NaBH_4$, 98%, Merck, China) was used as a $H_2$ source. Methanol (99.9%, Carlo Erba, France) was used as the reaction medium. Double distilled water (GFL 2108) was used for the required experiments.

### 2.2. Synthesis of g-C₃N₄ and Heteroatom-Doped g-C₃N₄ (H@g-C₃N₄)

The synthesis of the graphitic carbon nitride ($g\text{-}C_3N_4$) was carried out by heating the polymerization of the melamine by following the literature with some modifications [29–31]. In brief, 10 g of melamine was placed into a porcelain crucible and closed with a porcelain cover. After that, this porcelain crucible was placed into a muffle furnace and heated up to 550 °C with a heating rate of 3 °C/min. The melamine-contained porcelain crucible was kept at 550 °C for 4 h. Finally, the porcelain crucible was cooled to room temperature, and the obtained yellow solid was first pulverized in a mortar and then placed in 100 mL of water and sonicated 3 times for 30 min. The prepared $g\text{-}C_3N_4$ structures were centrifuged at 10,000 rpm and room temperature after each sonication step. Lastly, the prepared and washed $g\text{-}C_3N_4$ structures were dried with a freeze-dryer (Alpha 2-4 LSC, Christ), and stored in closed tubes for further usage.

On the other hand, for the synthesis of the heteroatom-doped $g\text{-}C_3N_4$ (H@g-$C_3N_4$, H: B, P or S) structures, the mentioned procedure was used. In short, melamine (80 mmol) and a 1:1 mole ratio of boric acid, phosphorus red, and sulfur, according to the melamine as B,

P, and S sources, was placed into a mortar and mechanically homogenized with physical mixing. After that, the mixtures were placed into porcelain crucibles separately, heated up to 550 °C with a 3 °C/min heating rate, and kept at 550 °C for 4 h. A similar procedure mentioned above was applied to the washing and drying of the prepared H@g-$C_3N_4$ (H: B, P, or S) structures. The washed and dried H@g-$C_3N_4$ (H: B, P, or S) structures were stored in closed tubes for further usage.

### 2.3. Instruments

The transmission electron microscopy (TEM, JEOL JEM-ARM200CFEG) images of the g-$C_3N_4$ structure were taken. The Fourier transform infrared (FT-IR, Spectrum, Perkin Elmer) spectroscopy and X-ray diffraction (XRD, PANalytical X'Pert Pro MPD) were used for the structural characterization of g-$C_3N_4$. A thermal gravimetric analyzer (TGA, SII TG/DTA6300, Exstar) was used for the determination of the thermal stabilities of the fluorescence spectroscopy (Lumina, Thermo) for the determination of the optical properties of the g-$C_3N_4$ structures.

### 2.4. Catalytic Activity of H@g-$C_3N_4$ on Dehydrogenation of $NaBH_4$ in Methanol

The catalytic activity of the prepared g-$C_3N_4$-based structures on the dehydrogenation of the $NaBH_4$ reaction in methanol was investigated by following earlier reported studies by our group [25,26]. In brief, 50 mg of g-$C_3N_4$-based structures were placed into a round bottom 50 mL flask as a catalyst; after that, the freshly weighed 0.0965 g $NaBH_4$ (2.55 mmol) was added into the flask. Finally, the 20 mL of methanol was quickly added into the flask attached to a homemade setup, and the produced hydrogen ($H_2$) was recorded as a function of time from the volumetric cylinder at 25 °C under continuous mixing at 1000 rpm. The homemade setup included a concentrated $H_2SO_4$-filled trap to catch the possible exhausted methanol moisture, and a water-filled and reversed volumetric cylinder to record the produced hydrogen as mL.

Moreover, the effect of the doping agent (B, P, and S) and temperature (−10, 0, 10, 25, and 40 °C) on the catalytic activity of the g-$C_3N_4$-based structures was investigated. The reusability of the prepared g-$C_3N_4$ and H@g-$C_3N_4$ (H: B, P, or S) structures were also investigated and compared with each other.

### 2.5. Calculation of Activation Parameters

The important activation parameters, such as activation energy (Ea), enthalpy (ΔH), and entropy (ΔS) values, were calculated with the application of the observed results from the g-$C_3N_4$-based-structures-catalyzed dehydrogenation of $NaBH_4$ in methanol at various temperatures mentioned above to the well-known Arrhenius (Equation (2)) and Eyring (Equation (3)) equations.

$$lnk = lnA - (Ea/RT) \tag{2}$$

$$ln\,(k/T) = -(\Delta H/R)(1/T) + ln(k_B/h) + \Delta S/R \tag{3}$$

where k is the reaction rate constant and was calculated according to a first-order kinetic expression [18,32], $E_a$ is the activation energy, T is the absolute temperature, $k_B$ is the Boltzmann constant ($1.381 \times 10^{-23}$ J.K$^{-1}$), h is the Planck's constant ($6.626 \times 10^{-34}$ J.s), ΔH is the activation enthalpy, ΔS is the entropy, and R is the gas constant (8.314 J.K$^{-1}$.mol$^{-1}$).

### 2.6. Reusability of g-$C_3N_4$-Based Structures on Dehydrogenation of $NaBH_4$ in Methanol

The reusability of the g-$C_3N_4$-based structures as catalysts for the dehydrogenation of $NaBH_4$ in methanol was investigated and compared to each other. The reusability of the catalysts on the reaction of the dehydrogenation of $NaBH_4$ in methanol was carried by the addition of 0.0965 g of $NaBH_4$ into a 50 mg catalyst containing 20 mL methanol for five consecutive times at 40 °C. There were two main parameters: activity% of the catalyst, and the conversion ability of the catalyst on reaction after consecutive usages on the dehydrogenation of $NaBH_4$ in methanol was determined. The activity% of the catalyst

was calculated from the calculated HGR values from half of the produced hydrogen, and the HGR value of the first usage was assumed as 100%. On the other hand, the conversion% is defined as the produced amount of hydrogen via the catalyzed reaction according to the stoichiometry of the dehydrogenation of $NaBH_4$ in methanol. After the initial dehydrogenation of $NaBH_4$ in the methanol reaction, a new $NaBH_4$ with the same amount, as was in the first use (0.0965 g), was added four more times, and the change in the activity% of the catalyst, and the conversion% of the reactions were determined for each use. All the reusability tests of the $g-C_3N_4$-based structures on the dehydrogenation of $NaBH_4$ in the methanol reactions were conducted in triplicates, and the results of the activity% of the catalyst and the conversion% of the reaction were presented with standard deviations.

## 3. Results and Discussion

### 3.1. Synthesis and Characterization of H@g-C₃N₄

Graphitic carbon nitride ($g-C_3N_4$) has a layered structure that is similar to that of graphite, and exhibits interesting and distinctive physicochemical properties because of the presence of s-triazine cores, despite the relatively low conductivity limiting its use in electronic and electrochemical processes [2,4,11,33]. The consensus is that the structure of $g-C_3N_4$ originates from molecules created by the direct coupling of the C-N, urea, and ethylenediamine with cyanamide, melamine, and their polymerized derivatives, which lead to either triazine-based or heptazine-based structures [1,2,8,10,33]. Based on heptazine, $g-C_3N_4$ and heteroatom-doped $g-C_3N_4$ structures are schematically depicted in Figure 1.

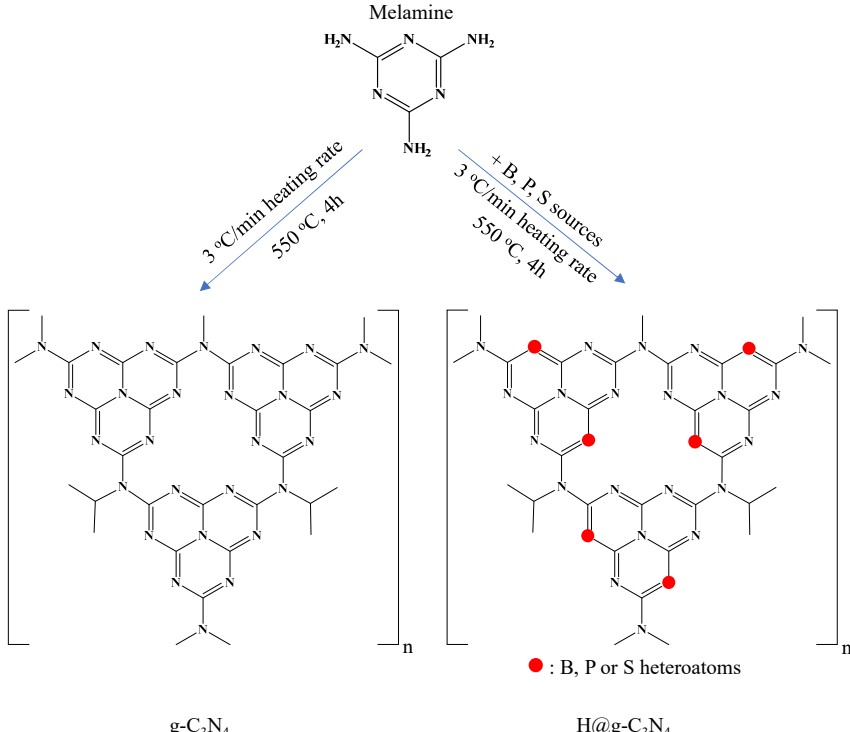

**Figure 1.** The schematic presentation of $g-C_3N_4$ and $H@g-C_3N_4$ structure.

Heteroatom-doping is supposed not to disturb the staking and in-layer structure of the networks and is reported as the heteroatoms participate in cyclic basic structural units of $g-C_3N_4$ structures [12,14–16]. It was also demonstrated as the heteroatoms B, P, and S were randomly distributed in structure.

The TEM images of the native $g-C_3N_4$ structures were also given in Figure 2a, and the corresponding XRD pattern of the native $g-C_3N_4$ was given in Figure 2b. The exact periodic units in each layer of $g-C_3N_4$ could be readily identified by the XRD peak associated with an in-plane structural packing. The typical experimental XRD pattern of the bulk $g-C_3N_4$ had two distinct diffraction peaks located at 27.40 and 13.4° 2Θ, which can be indexed

as (002) and (100) diffraction planes for graphic materials. The XRD results indicate that the g-C$_3$N$_4$ exhibited a flake-like structure with an interplanar stacking distance of 0.356 nm revealed by (002) diffraction. The g-C$_3$N$_4$ structures exhibited one distinct XRD diffraction peak at 17.4° 2Θ. This structure indicated the formation of the s-triazine units in the tubular g-C$_3$N$_4$.

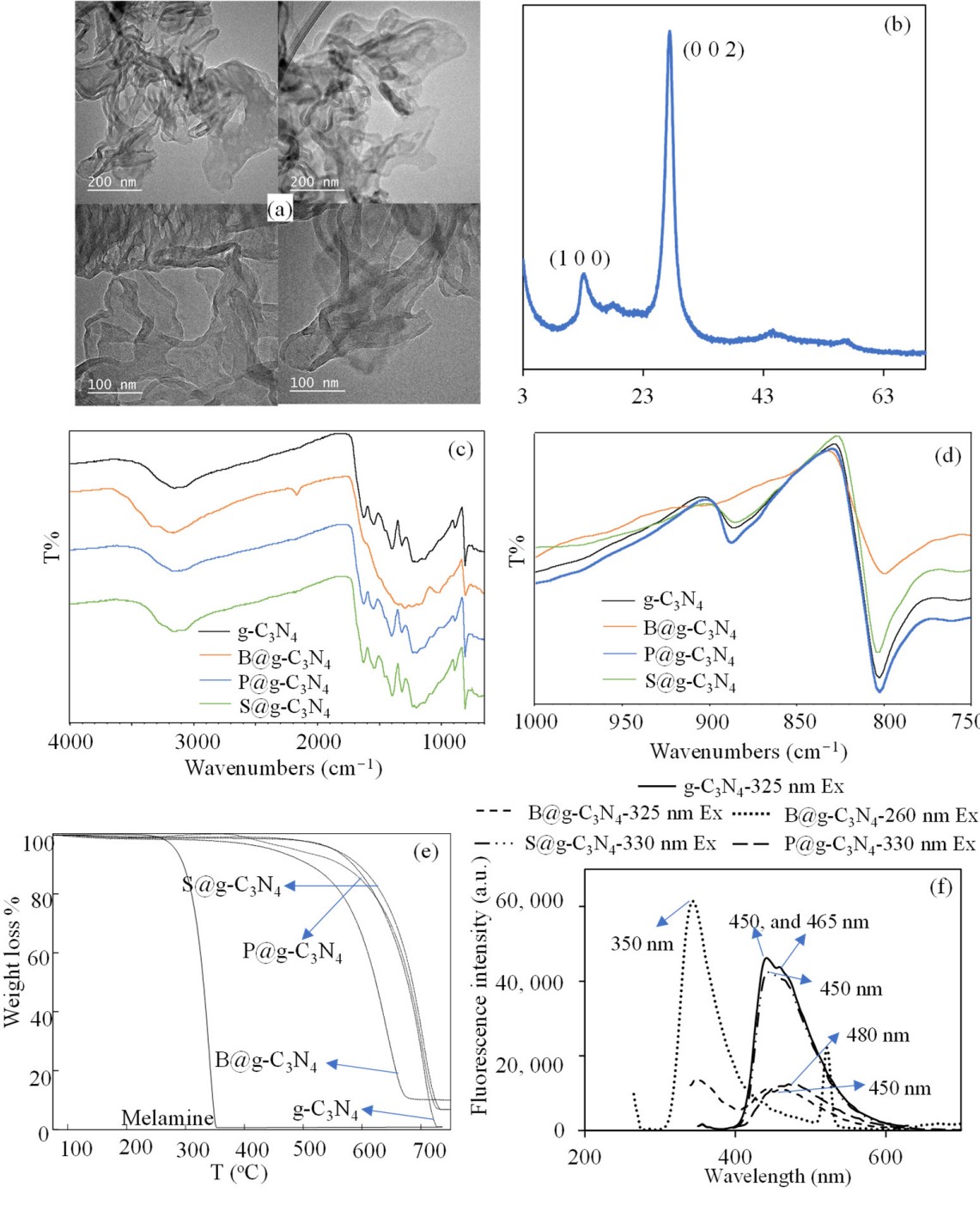

**Figure 2.** The (**a**) TEM images, (**b**) XRD pattern, (**c**) FT-R spectra, (**d**) detailed FT-IR spectra, (**e**) TGA thermogram, and (**f**) fluorescence spectra of g-C3N4-based structures.

On the other hand, the FT-IR spectra and TGA thermograms of the native g-C$_3$N$_4$ and H@g-C$_3$N$_4$ (H: B, P or S) are given in Figure 2c–e, respectively. The native g-C$_3$N$_4$ and the heteroatom-doped g-C$_3$N$_4$ structures exhibited matching spectra, as shown in Figure 2c. It is apparent that there was an increase or decrease in peak intensities for the heteroatom-

doped g-C$_3$N$_4$, particularly in the 750–1000 cm$^{-1}$ region, as illustrated in Figure 2d. It is more evidently seen in Figure 2d that the peak intensities of the heteroatom-doped g-C$_3$N$_4$, depending on the type of the heteroatoms, are increased or decreased in peak intensities in the 800–802 cm$^{-1}$ range. In both spectra, there were bands in the 1200–1600 cm$^{-1}$ range connected to the usual stretch modes of the aromatic C–N heterocycles [15,16]. At 807 cm$^{-1}$, the typical vibration of the triazine units was obtained [34]. At 3000–3500 cm$^{-1}$, terminal amino groups and surface-adsorbed OH bands were observed [35]. The peaks connected to the phosphorus groups in the g-C$_3$N$_4$ structure could not be seen in the FT-IR spectrum, either because there were insufficient heteroatoms or an overlapping of C-N bond vibration. There were also no significant changes in the TGA thermograms of the g-C$_3$N$_4$ and H@g-C$_3$N$_4$ (H: B, P, or S) structures in Figure 2e. The TGA analyses were carried out in the presence of 20 mL/min N$_2$ flow and heated up to 750 °C with a 10 °C/min heating rate. The melamine thermally degraded between 300 and 370 °C with more than 99% weight loss. On the other hand, the g-C$_3$N$_4$ and H@g-C$_3$N$_4$ (H: B, P, or S) structures were thermally stable up to almost 500 °C due to the synthesis of their structures at 550 °C and the degradation between 525 and 700 °C.

To confirm the doping of g-C$_3$N$_4$ with B, S, or P heteroatoms, the fluorescence properties of the structure were compared in Figure 2f. It was observed that g-C$_3$N$_4$ exhibited two emission wavelengths at 450 and 465 nm more than a 42,000 fluorescence intensity at 325 nm of the excitation wavelength. On the other hand, B@g-C$_3$N$_4$ excited at the 260 and 325 nm wavelength and exhibited emissions at 350 and 460 nm with a 60,000 and 11,500 fluorescence intensity, respectively. In addition, the excitation wavelengths for the P@g-C$_3$N$_4$ and S@g-C$_3$N$_4$ structures were also determined as 330 nm, and the emission wavelengths were obtained at 480 and 450 nm with fluorescence intensities of 12,000 and 40,000, respectively. The observed results from fluorescence spectrometers confirmed the successful doping of the g-C$_3$N$_4$ structures with B, P, and S heteroatoms, separately.

### 3.2. The Usage of g-C$_3$N$_4$-Based Structures as a Catalyst on Dehydrogenation of NaBH$_4$ in Methanol

A handmade setup was utilized to determine the catalytic activity of the g-C$_3$N$_4$-based structures on the dehydrogenation of NaBH$_4$ in methanol. A Round bottom flask (50 mL) containing catalyst and NaBH$_4$ was filled with 20 mL of methanol. This bottle was soon connected to the trap carrying concentrated sulfuric acid, which was also connected to the water-filled and inverted volumetric cylinder. The hydrogen generated in the flask was passed from the trap to collect any methanol moisture and then to the water-filled volumetric cylinder in this setup. The generated H$_2$ was replaced with water in the cylinder, and its volume was calculated using the volumetric cylinder.

### 3.2.1. The Effect of Heteroatom Doping on the Catalytic Activity of g-C$_3$N$_4$

The dehydrogenation of NaBH$_4$ in methanol has several benefits over the dehydrogenation of NaBH$_4$ in water, including faster reaction rates, metal-free catalysis, functioning at subzero temperatures, and so on [21,22,25]. As a result, the potential catalytic activity of the produced g-C$_3$N$_4$-based structures on NaBH$_4$ dehydrogenation in methanol was evaluated. The comparison of reaction rate (mL/min) as a function of time for the self-methanolysis and the g-C$_3$N$_4$-based-structures-catalyzed dehydrogenation of NaBH$_4$ in methanol was given in Figure S1. As can be seen from Figure 3a, the self- and g-C$_3$N$_4$-catalyzed dehydrogenation of NaBH$_4$ in methanol took 32.5 min and produced 250 ± 2 mL of H$_2$. It can be stated that there was no catalytic activity of g-C$_3$N$_4$ on the dehydrogenation of NaBH$_4$ in methanol.

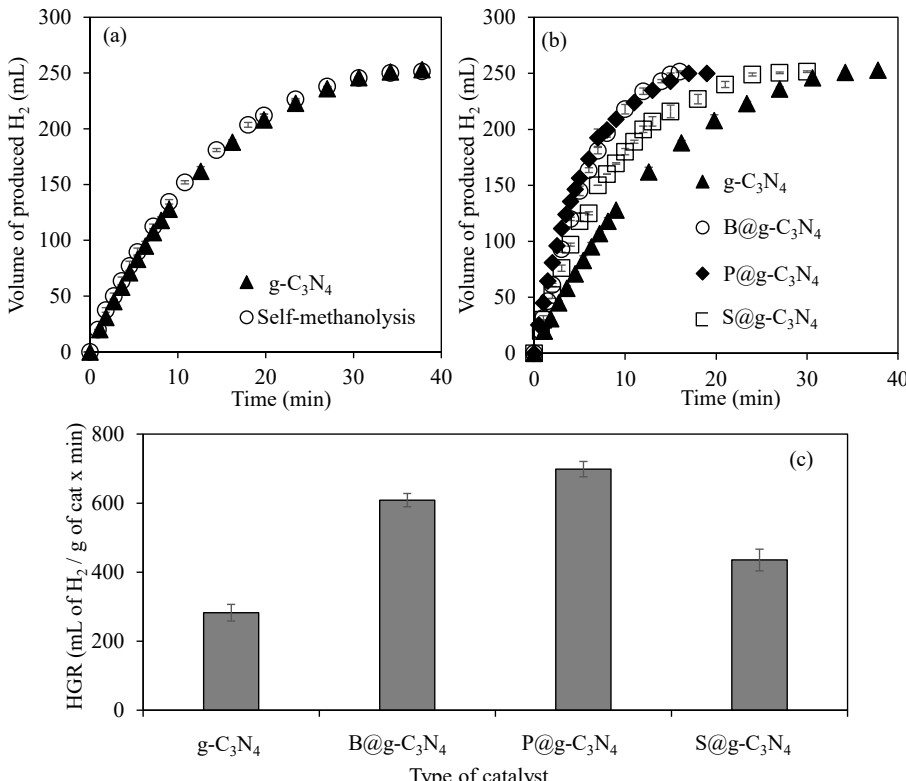

**Figure 3.** The comparison of catalytic activity of (**a**) native g-C$_3$N$_4$ with self-methanolysis, (**b**) native g-C$_3$N$_4$ with H@g-C$_3$N$_4$ (H: B, P, or S) on the dehydrogenation reaction of NaBH$_4$ in methanol, and the comparison of calculated HGR values for the g-C$_3$N$_4$-based-structures-catalyzed dehydrogenation reaction of NaBH$_4$ in methanol [reaction condition: 50 mg catalyst, 20 mL methanol, 0.0965 g NaBH$_4$, 25 °C, 1000 rpm].

On the other hand, the prepared heteroatom-doped B@g-C$_3$N$_4$, P@g-C$_3$N$_4$, and S@g-C$_3$N$_4$ structures catalyzed the dehydrogenation of NaBH$_4$ in methanol. In Figure 3b, the reaction was completed with the production of 250 ± 2 mL of H$_2$ under B@g-C$_3$N$_4$, P@g-C$_3$N$_4$, and S@g-C$_3$N$_4$ catalyzed at 17, 19, and 30 min, respectively. The calculated hydrogen generation rate (HGR, mL H$_2$/g of cat x min) values for the g-C$_3$N$_4$-, B@g-C$_3$N$_4$-, P@g-C$_3$N$_4$-, and S@g-C$_3$N$_4$-catalyzed dehydrogenation of NaBH$_4$ in methanol are compared in Figure 3c. It was observed that the heteroatom-doped g-C$_3$N$_4$-structures-catalyzed reactions were faster than the native g-C$_3$N$_4$-structures-catalyzed dehydrogenation of NaBH$_4$ in methanol. On the other hand, the P@g-C$_3$N$_4$-catalyzed dehydrogenation of NaBH$_4$ in methanol with 699 ± 22 mL H$_2$/g of cat x min HGR values was faster than both B@g-C$_3$N$_4$- and S@g-C$_3$N$_4$-catalyzed reactions with HGR values of 609 ± 19 and 435 ± 31 mL H$_2$/g of cat x min, respectively.

3.2.2. The Effect of Temperature on g-C$_3$N$_4$-Based-Structures-Catalyzed Dehydrogenation of NaBH$_4$ in Methanol

The effect of the reaction temperature on the catalytic activity of the g-C$_3$N$_4$-based structures was also investigated by carrying out reactions at various temperatures between −10 and 40 °C. The reaction rate (mL/min) vs. time graphs of H@g-C$_3$N$_4$ catalyzed dehydrogenation of the NaBH$_4$ reaction were given in Figure S2. As seen from Figure 4a, the B@g-C$_3$N$_4$ catalyzed reactions at 10, 25, and 40 °C were completed in 48, 16, and 7 min, with the production of 250 ± 2 mL of H$_2$. It was also observed that 200 ± 5 mL of H$_2$ was produced from the dehydrogenation of NaBH$_4$ in methanol, even at 0 and −10 °C in 32.5 and 51 min, respectively, in the presence of under B@g-C$_3$N$_4$ as a catalyst. The calculated HGR values are also compared in Figure 4b, and it was observed that increasing



the temperature from −10 to 40 °C increased the HGR values almost 10 times from $125 \pm 8$ to $1346 \pm 59$ mL $H_2$/g of cat x min.

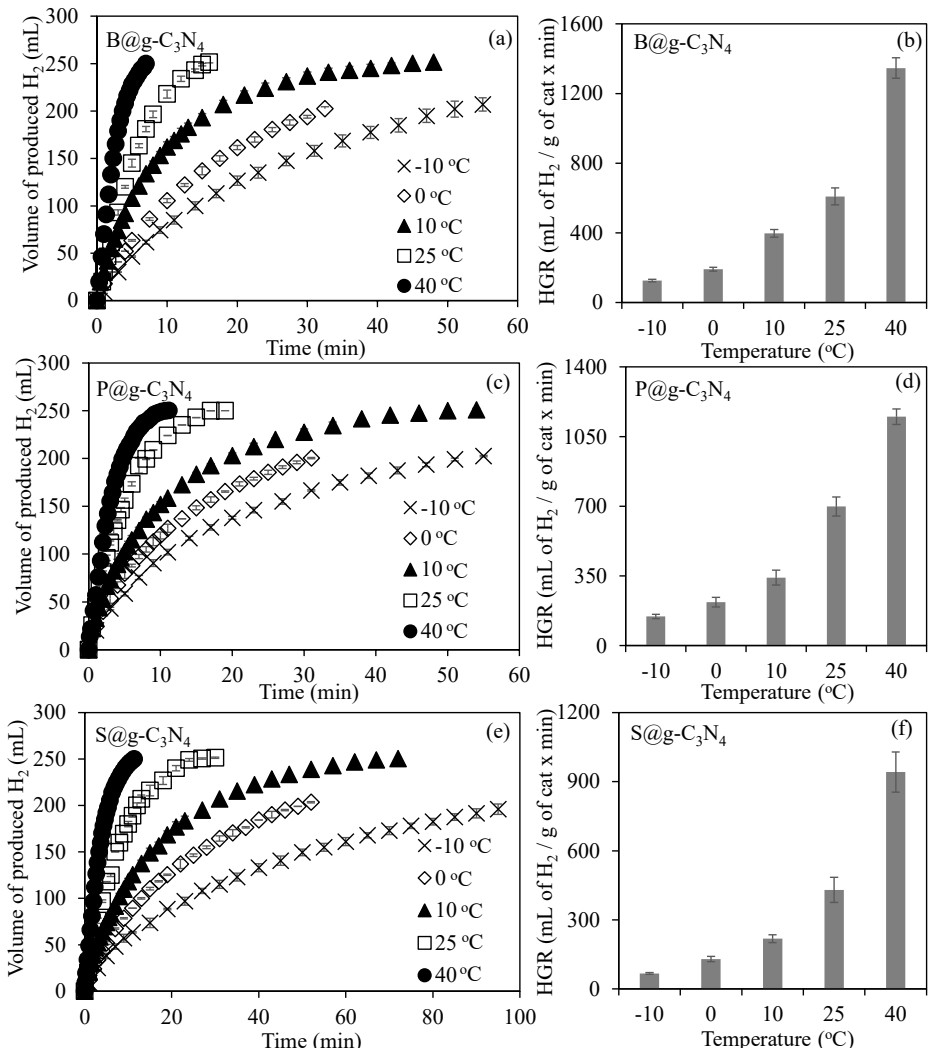

**Figure 4.** (**a**) The effect of temperature on catalytic activity of B@g-$C_3N_4$, and (**b**) comparison of calculated HGR values for the B@g-$C_3N_4$-catalyzed dehydrogenation reaction of NaBH$_4$ in methanol, (**c**) the effect of temperature on catalytic activity of P@g-$C_3N_4$, and (**d**) comparison of calculated HGR values for the P@g-$C_3N_4$-catalyzed dehydrogenation reaction of NaBH$_4$ in methanol, (**e**) the effect of temperature on catalytic activity of S@g-$C_3N_4$, and (**f**) comparison of calculated HGR values for the S@g-$C_3N_4$-catalyzed dehydrogenation reaction of NaBH$_4$ in methanol at various temperatures [reaction condition: 50 mg catalyst, 20 mL methanol, 0.0965 g NaBH$_4$, 1000 rpm].

Similarly, the reaction rates of the P@g-$C_3N_4$-catalyzed dehydrogenation of NaBH$_4$ in methanol increased with the increasing temperature, and the corresponding graph is given in Figure 4c. It was revealed that $200 \pm 5$ mL of $H_2$ was generated from the P@g-$C_3N_4$-catalyzed dehydrogenation of NaBH$_4$ in methanol in 55, 31, 20, 8, and 5 min at −10, 0, 10, 25, and 40 °C, respectively. Additionally, $250 \pm 2$ mL of $H_2$ was produced in 54, 17, and 11 min at 10, 25, and 40 °C from the dehydrogenation of NaBH$_4$ in methanol in the presence of P@g-$C_3N_4$ as a catalyst. In Figure 4d, the calculated HGR values for the P@g-$C_3N_4$-catalyzed reaction at various temperatures are shown, and the HGR values increased approximately eight times from −10 to 40 °C with $146 \pm 8$ to $1150 \pm 40$ mL $H_2$/g of cat x min.

The effect of temperature on the S@g-$C_3N_4$-catalyzed dehydrogenation reaction of NaBH$_4$ in methanol is given in Figure 4e. The dehydrogenation reaction of NaBH$_4$ in

methanol was completed in 72, 27, and 11 min at 10, 25, and 40 °C with 250 ± 2 mL of $H_2$ production in the presence of S@g-$C_3N_4$ as a catalyst. Moreover, even at 0 and −10 °C, 200 ± 5 mL of $H_2$ production was achieved in 52 and 95 min from the S@g-$C_3N_4$-catalyzed dehydrogenation reaction of $NaBH_4$ in methanol, respectively. The HGR values calculated for the S@g-$C_3N_4$-catalyzed dehydrogenation reaction of $NaBH_4$ in methanol at various temperatures were compared in Figure 4f. It was determined that the HGR values increased approximately 15 times with the increasing of temperature from −0 to 40 °C as 67 ± 4 to 942 ± 87 mL $H_2$/g of cat x min.

Although the rates of the reaction catalyzed by the g-$C_3N_4$-based structures increased with the increase in temperature, it was observed that the reactions catalyzed by B@g-$C_3N_4$ and P@g-$C_3N_4$ were faster than the reactions catalyzed by S@g-$C_3N_4$.

### 3.3. Comparison of Activation Parameters for g-$C_3N_4$-Based-Structures-Catalyzed Reaction

The calculated activation parameters, such as Ea, $\Delta H$, and $\Delta S$ values, for the g-$C_3N_4$-based-structures-catalyzed dehydrogenation reaction of $NaBH_4$ in methanol from Figure 4 via Arrhenius and Eyring equations are summarized in Table 1. In Figure S3, the corresponding Arrhenius and Eyring plots of the g-$C_3N_4$-based-structures-catalyzed dehydrogenation reaction of $NaBH_4$ in methanol were given.

**Table 1.** The calculated Ea, $\Delta H$, and $\Delta S$ values for H@g-$C_3N_4$-catalyzed (H: B, P, or S) dehydrogenation reactions of $NaBH_4$ in methanol and in comparison with reported studies.

| Materials | Activation Parameters | | | Ref. |
|---|---|---|---|---|
| | Ea (kJ/mol) | $\Delta H$ (kJ/mol) | $\Delta S$ (J/mol.K) | |
| Self methanolysis | 52.9 | - | - | [36] |
| | 62.9 | | | [18] |
| B@g-$C_3N_4$ | 31.2 | 28.2 | −189.1 | This study |
| P@g-$C_3N_4$ | 26.9 | 24.0 | −191.6 | |
| S@g-$C_3N_4$ | 31.2 | 28.2 | −191.5 | |
| O doped g-$C_3N_4$ | 36.1 | - | - | [15] |
| P doped g-$C_3N_4$ | 30.3 | - | - | [16] |
| Co-P/CNTs-Ni foam | 49.9 | - | - | [37] |
| Ru–Co/C | 36.8 | - | - | [38] |
| CS from lactose | 23.8 | 21.4 | −173 | [25] |
| Metal-free OP-$H_3PO_4$-Cat | 12.5 | - | - | [39] |

It was determined that the Ea values for the B@g-$C_3N_4$-, P@g-$C_3N_4$-, and S@g-$C_3N_4$-catalyzed dehydrogenation of $NaBH_4$ in methanol as 31.2, 26.9, and 31.2 kJ/mol, respectively. The Ea values for the non-catalyzed dehydrogenation of $NaBH_4$ in methanol were reported as 52.9 kJ/mol [36], and 62.9 kJ/mol [18], separately. Moreover, the Ea values for oxygen (O) and phosphorus (P)-doped g-$C_3N_4$-catalyzed dehydrogenation of $NaBH_4$ in methanol were reported as 36.1 kJ/mol [15] and 30.3 kJ/mol [16], respectively. Also, different catalysts reported a wide range of Ea values, for example, the Ea value of 49.9 kJ/mol for Co-P nanoparticles supported on dandelion-like CNTs-Ni foam composites [37], 36.8 kJ/mol for RuCo bimetallic nanoparticles supported carbon black [38], 23.8 kJ/mol for carbon spheres catalayst from lactose [25], and 12.47 kJ/mol for the orange peel waste protonated with a phosphoric acid-catalyzed reaction [39].

In the comparison of calculated Ea values for the B@g-$C_3N_4$-, P@g-$C_3N_4$-, and S@g-$C_3N_4$-catalyzed dehydrogenation of $NaBH_4$ in methanol with reported studies in the literature, it can be clearly stated that the determined Ea values are lower and comparable with reported studies about carbon-based catalysts.

### 3.4. Reusabilities of g-C₃N₄-Based Catalysts on Dehydrogenation of NaBH₄ in Methanol

In addition to being environmentally friendly, repetitive usages of catalysts are a very important parameter for industrial applications to keep the cost low. Therefore, catalysts that can be used repeatedly with high activity without loss of activity are very important for industrial applications. The graphs examining the reusability of heteroatom-doped g-C₃N₄-based catalysts in the dehydrogenation reaction of $NaBH_4$ in methanol are given in Figure 5. In Figure 5a, the changing in the conversion% of a reaction, and the activity% of B@g-C₃N₄ structures were investigated for the B@g-C₃N₄-catalyzed dehydrogenation reaction of $NaBH_4$ in methanol for five consecutive usages of the catalyst. It was observed that 100% conversion for B@g-C₃N₄ catalyzed the dehydrogenation reaction of $NaBH_4$ in methanol at each usage of the catalyst. On the other hand, the activity of the B@g-C₃N₄ catalyst in reaction decreased to 87 ± 1% after five consecutive uses.

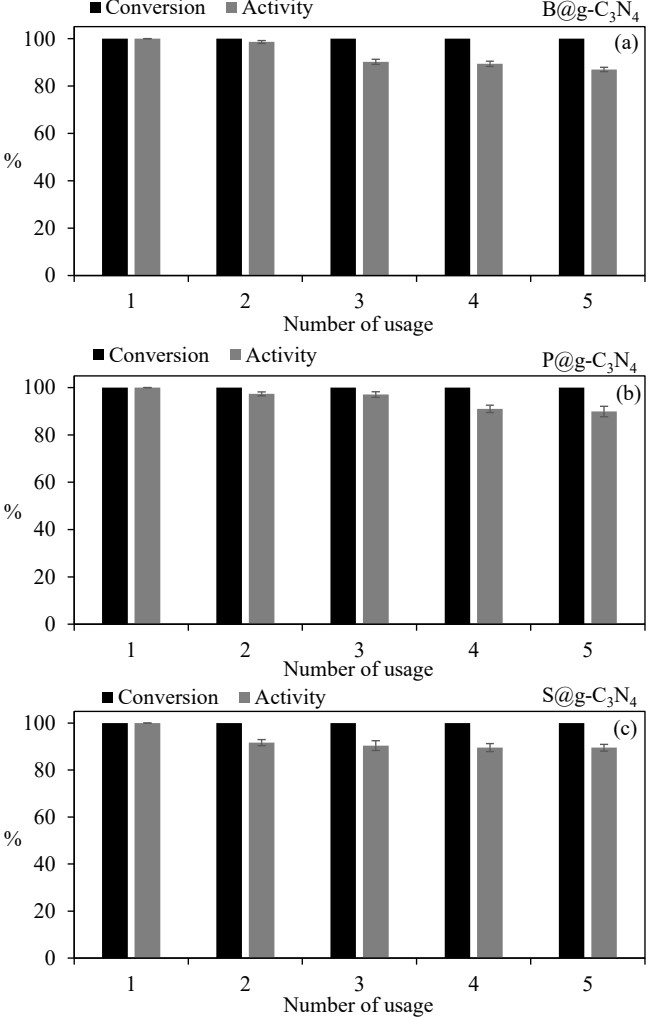

**Figure 5.** The reuseability of (**a**) B@g-C₃N₄, (**b**) P@g-C₃N₄, and (**c**) S@g-C₃N₄ catalysts in dehydrogenation reaction of $NaBH_4$ in methanol [reaction condition: 50 mg catalyst, 20 mL methanol, 0.0965 g $NaBH_4$, 40 °C, 1000 rpm].

In Figure 5b, the conversion% of the P@g-C₃N₄-catalyzed dehydrogenation reaction of $NaBH_4$ in methanol was also observed as 100% for every five usages. Moreover, the P@g-C₃N₄ structures catalyzed the dehydrogenation reaction of $NaBH_4$ in methanol with almost 100% activity for three consecutive usages, and the activity% of the catalyst decreased to 90 ± 2% after the fifth usage in a row. Similarly, the exhibited 100% conversion and activity% decreasing after the five consecutive usages of the S@g-C₃N₄ catalysts on the dehydrogenation reaction of $NaBH_4$ in methanol were also illustrated in Figure 5c. Overall,

all g-$C_3N_4$-based-structures-catalyzed dehydrogenation reaction of $NaBH_4$ in methanol was completed with 100% conversion for each usage. However, the activity% of the catalyst decreased with five consecutive usages. The decrease in the activity% of all g-$C_3N_4$-based structures after consecutive usages can be explained by the accumulation of the byproduct of the dehydrogenation reaction of $NaBH_4$ in methanol [$B(OCH_3)_4$] on the active sides of catalysts [40].

## 4. Conclusions

The preparation of H@g-$C_3N_4$ (H: B, P, or S) structures as a metal-free catalyst for the dehydrogenation reaction of $NaBH_4$ in methanol was reported. The prepared heteroatom-doped B@g-$C_3N_4$, P@g-$C_3N_4$, and S@g-$C_3N_4$ structures catalyzed the dehydrogenation reaction of $NaBH_4$ in methanol faster than the native g-$C_3N_4$ structures, with almost 2-, 2.5-, and 1.5-fold higher reaction rates. Among the B-, P-, and S-doped g-$C_3N_4$ structures-catalyzed-reactions, the P@g-$C_3N_4$-catalyzed one showed the best catalytic activity at 25 °C with 699 $\pm$ 49 mL $H_2$/g of cat.min. This work shares the drawbacks encountered in actual applications due to the lack of active sites of bare g-$C_3N_4$ [16]. However, doping with metal-free heteroatoms, such as B, S, and P, to improve the properties of bare g-$C_3N_4$, such as the electrical, functional, and textural characteristics, is proven to overcome these difficulties [41,42]. More active sites may be produced by doping heteroatoms into carbonaceous structures, which can improve their catalytic activity by endowing variations in electron density, bond lengths, and atomic sizes depending on the doping agents [43]. Among these heteroatoms, doping P, which has a higher covalent radius than both B and S and has an electronegativity between B and S, can boost the catalytic activity of g-$C_3N_4$ more effectively [44]. Moreover, the calculated Ea values for the B@g-$C_3N_4$-, P@g-$C_3N_4$-, and S@g-$C_3N_4$-structures-catalyzed dehydrogenation reactions of $NaBH_4$ in methanol with 31.2, 26.9, and 31.2 kJ/mol were remarkable, where Ea values calculated for the non-catalyzed self-methanolysis reaction was between 52.9 and 62.9 kJ/mol. Moreover, the repetitive usages of the prepared B@g-$C_3N_4$, P@g-$C_3N_4$, and S@g-$C_3N_4$ structures as catalysts on the dehydrogenation reaction of NaBH4 in methanol showed that the catalysts provided more than 80% activity even after five consecutive usages, with 100% conversion for each. Aside from the ability to customize the features and functionality of the g-$C_3N_4$-based catalyst for the design of the next generation of $H_2$ production technologies, the ability of these $H_2$ generation systems to operate even at lower temperatures, e.g., below 0 °C using $NaBH_4$ in methanol, is an important aspect to the $H_2$ energized applications in colder climates, and enhancing the use of these kinds of systems for the upcoming generation future application potentials.

**Supplementary Materials:** The following supporting information can be downloaded at: https://www.mdpi.com/article/10.3390/c8040053/s1, Figure S1: The comparison of reaction rate vs time plots of (a) native g-$C_3N_4$ with self-methanolysis, (b) native g-$C_3N_4$ with H@g-$C_3N_4$ (H: B, P or S) on dehydrogenation reaction of $NaBH_4$ in methanol [Reaction condition: 50 mg catalyst, 20 mL methanol, 0.0965 g $NaBH_4$, 25 °C, 1000 rpm]. Figure S2: The comparison of calculated reaction values for (a) B@g-$C_3N_4$, (b) P@g-$C_3N_4$, and (c)S@g-$C_3N_4$ catalyzed dehydrogenation reaction of $NaBH_4$ in methanol. Figure S3. The plotted (a) Arrhenius, and (b) Eyring graph for B@g-$C_3N_4$ catalyzed reaction, (c) Arrhenius, and (d) Eyring graph for P@g-$C_3N_4$ catalyzed reaction, and (e) Arrhenius, and (f) Eyring graph for S@g-$C_3N_4$ catalyzed reaction.

**Author Contributions:** Conceptualization, N.S.; methodology, S.D. and N.S.; validation, S.D.; formal analysis, S.D. and N.S.; investigation, S.D. and N.S.; resources, N.S.; data curation, S.D. and N.S.; writing—original draft preparation, S.D.; writing—review and editing, N.S.; visualization, N.S.; supervision, N.S.; project administration, N.S.; funding acquisition, N.S. All authors have read and agreed to the published version of the manuscript.

**Funding:** This research received no external funding.

**Data Availability Statement:** The data presented in this study are available on request from the corresponding author.

**Acknowledgments:** The authors gratefully acknowledge the financial support provided by the Scientific Research Commission of Canakkale Onsekiz Mart University (FBA-2022-4183).

**Conflicts of Interest:** The authors declare no conflict of interest.

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
