# Peer review of "Use of Heteroatom-Doped g-C3N4 Particles as Catalysts for Dehydrogenation of Sodium Borohydride in Methanol"

_carbon, 2020_

Round 1

Reviewer 1 Report

1. English should be revised for minor errors.

2. In line 132 you wrote hydrogel, is that correct?

3. In the section 2.5 you calculate the activation parameters by considering the 0 reaction order. However, you do not provide any reason for such a simplification. Please include an explanation about this decision.

4. Why hydrogen is analyzed ex-situ? How you confirmed the reliability of the measurements?

5. You argued that the peak intensities in the FTIR between 750 - 1000 were increased. This is not shown in Fig. 2c. In addition, to compare these spectrum you must normalize them. Please explain the signal processing procedure that you used to obtain these conclusions.

6. I think all the catalytic data should be provided in reaction rate units instead volumetric production of hydrogen. In addition, the results shpuld be compared to literature using catalyst-related properties.

Author Response

Reviewer 1

Comments and Suggestions for Authors

  1. English should be revised for minor errors.

- The manuscript was re-checked and the English level of the manuscript is improved significantly.

  1. In line 132 you wrote hydrogel, is that correct?

- We are sorry for the mistake, yt was corrected as “hydrogen”.

  1. In the section 2.5 you calculate the activation parameters by considering the 0 reaction order. However, you do not provide any reason for such a simplification. Please include an explanation about this decision.

- We apologize for this simple mistake, and this was corrected in the revised manuscript the following information is provided on pp 3 of the revised manuscript as;

“... where k is the reaction rate constant and was calculated according to a first-order kinetic expression [18,32],...”.

[18] Lo, C.T.F.; Karan, K.; Davis, B.R. Kinetic Studies of Reaction between Sodium Borohydride and Methanol, Water, and Their Mixtures. Ind. Eng. Chem. Res. 2007, 46, 5478–5484, doi:10.1021/ie0608861.

[32] Demirci, S.; Ari, B.; Butun Sengel, S.; Ä°nger, E.; Sahiner, N. Boric acid versus boron trioxide as catalysts for green energy source H2 production from sodium borohydride methanolysis. MANAS J. Eng. 2021, doi:10.51354/mjen.980286.

  1. Why hydrogen is analyzed ex-situ? How you confirmed the reliability of the measurements?

- In literature, there have been many studies reported using this well-established method by our group and some other research groups about the methanolysis of NaBH4 reactions to produce hydrogen as the reliable measurements for this reaction mechanism [15,16,18, 22, 25, 26, 32]. Therefore, in our opinion, the used method in this manuscript is the most reliable method for the measurement of the produced H2.

  1. You argued that the peak intensities in the FTIR between 750 - 1000 were increased. This is not shown in Fig. 2c. In addition, to compare these spectrum you must normalize them. Please explain the signal processing procedure that you used to obtain these conclusions.

- The following information is given on pp 6 of the revised manuscript as;

“It is apparent that there was an increase or decrease in peak intensities for the heteroatom-doped g-C3N4, particularly in the 750-1000 cm-1 region as illustrated in Figure 2(d). It is more evidently seen in Figure 2(d) that the peak intensities of the heteroatom-doped g-C3N depending on the type of the heteroatoms are increased or decreased in peak intensities in the 800-802 cm-1 range.”

  1. I think all the catalytic data should be provided in reaction rate units instead of volumetric production of hydrogen. In addition, the results should be compared to literature using catalyst-related properties.

- The comparison of the data with the reported other studies, the volume of produced H2 was kept in the related Figures. On the other hand, the reaction rate (mL/min) vs time (min) plots for the corresponding graphs of Figure 3 and 4 were added to Supporting Figures as Figure S1, and S2. The comparison of corresponding hydrogen generation rate (HGR) values was also given in both Figure 3(c) and Figure 4(b), (d), and (f). The following information is provided on pp 6 of the revised manuscripts as;

“The comparison of reaction rate (mL/min) as a function of time for the self-methanolysis and by g-C3N4 based structures catalyzed dehydrogenation of NaBH4 in methanol was given in Figure S1.

and on pp 7 of the revised manuscript as;

“The reaction rate (mL/min) vs time graphs of H@g-C3N4 catalyzed dehydrogenation of NaBH4 reaction were given in Figure S2.”

Reviewer 2 Report

This manuscript could be accepted for publication after minor revision.

 1.         What is the meaning of “increased the catalytic effect” in the following sentence? “The doping of g-C3N4 structures with heteroatoms increased the catalytic effect of the native g-C3N4 structure.”

2.         Why use “However” in this position. “However, g-C3N4 structures have also been reported to act as a highly efficient and photostable organic photocatalyst after doping with heteroatoms such as S, B, O, and P.”

3.         The descriptions of “little conversion ability” and “slow reaction kinetics” feel strange in the sentence of “Since the NaBH4 hydrolysis reaction has little conversion ability and slow reaction kinetics at low temperature”.

4.         About the blank reaction (without catalyst), the effect of reaction temperature on the self-catalyzed dehydrogenation should be also investigated.

5.         By combining with the related literatures, more discussion about the catalytic active sites and reaction mechanism should be provided.

6.         Some typos and grammar mistakes are present, for instance, Line 13, graphitic carbon nitride (g-C3N4) from melamine as precursor; Line 17, The prepared g-C3N4-based structures were used as a catalyst; Line 197, or the CN bond vibration overlapped.

Author Response

Reviewer 2

Comments and Suggestions for Authors

This manuscript could be accepted for publication after minor revision.

-We thank the reviewers for their time and efforts and nice comments to contribute to the quality of of the manuscript. We thank the reviewers’ suggestions point-by-point and improve the manuscript significantly;

  1. What is the meaning of “increased the catalytic effect” in the following sentence? “The doping of g-C3N4 structures with heteroatoms increased the catalytic effect of the native g-C3N4 structure.”

-The sentence was rewritten as;

 “The catalytic performance of H@g-C3N4 (H=B, S or P) structures in the dehydrogenation reaction of sodium borohydride (NaBH4) in methanol was determined higher than the catalytic performance of the bare g-C3N4 structure.”

  1. Why use “However” in this position. “However, g-C3N4 structures have also been reported to act as a highly efficient and photostable organic photocatalyst after doping with heteroatoms such as S, B, O, and P.”

-It was removed. (pls see pp 1, line 42-43 of the revised manuscript.)

  1. The descriptions of “little conversion ability” and “slow reaction kinetics” feel strange in the sentence of “Since the NaBH4 hydrolysis reaction has little conversion ability and slow reaction kinetics at low temperature”.

- The sentence was revised on pp 2 of the revised manuscript as;

“Since the NaBH4 hydrolysis reaction has low conversion rates and reaction kinetics at low temperatures, researchers have focused on finding new solvents for H2 production from NaBH4 using different solvents [17–19].”

  1. About the blank reaction (without catalyst), the effect of reaction temperature on the self-catalyzed dehydrogenation should be also investigated.

- The effect of reaction temperature on self-catalyzed dehydrogenation was reported many times in the literature and the corresponding Ea was calculated for the self-catalyzed dehydrogenation of NaBH4, and it was also cited in the revised manuscript as the references # [18, 36].

  1. By combining with the related literatures, more discussion about the catalytic active sites and reaction mechanism should be provided.

-The following information on pp 11 of the revised manuscript is given as;

“This work shares the drawbacks encountered in actual applications due to the lack of active sites of bare g-C3N4 [16]. However, by doping with metal-free heteroatoms such as B, S, and P to improve the properties of bare g-C3N4 such as the electrical, functional, and textural characteristics has been proven to overcome these difficulties [41,42]. More active sites may be produced by doping heteroatoms into carbonaceous structures which can improve their catalytic activity by endowing variations in electron density, bond lengths, and atomic sizes depending on the doping agents [43]. Among these heteroatoms, doping P, which has a higher covalent radius than both B and S and has an electronegativity between B and S, can boost the catalytic activity of g-C3N4 more effectively [44].”

  1. Some typos and grammar mistakes are present, for instance, Line 13, graphitic carbon nitride (g-C3N4) from melamine as precursor; Line 17, The prepared g-C3N4-based structures were used as a catalyst; Line 197, or the CN bond vibration overlapped.

-They are all re-checked and corrected as suggested.

Reviewer 3 Report

In this study, the authors have discussed the catalytic activity of heteroatom doped g-C3N4 particles. They found that B, S, and P doped g-C3N4 can generate more H2 than the undoped g-C3N4 in dehydrogenation of NaBH4 in methanol. In particular, the P doped g-C3N4 showed the best catalytic performance.

  Although the authors discovered interesting results, this paper does not involve fundamental analysis to discuss the catalytic performances. The authors did not chemically specify the catalytic active sites in this dehydrogenation reaction and did not provide quantitative analysis regarding the catalytic active sites in each sample. In this case, it is not possible to do fair-comparison of catalytic performances between different samples. Therefore, I suggest to reject this paper. Please see the more detailed comments below.

1)    How much heteroatoms can be incorporated in your synthetic methods? Specify the quantity (e.g. mol %) of heteroatoms in each g-C3N4 catalyst. 

2)    Clearly specify and prove the catalytic active sites in this dehydrogenation reaction. Since the undoped-gC3N4 also showed some catalytic performances, the newly added heteroatoms are not just a sole factor to cause this dehydrogenation reaction.

3)    Compare the turnover frequency of samples by estimating all of their catalytic active sites. By doing so, you will be able to quantitatively compare their catalytic performances.

4)    Prove that the collected gases are pure H2. (by using GC or mass spectrometer)

5)    Suggest how these heteroatoms can enhance the catalytic performances. In this paper, you just listed numbers proving the reaction results, but you did not provide your insights or proofs for the chemical and physical effects of heteroatoms.

Author Response

Reviewer 3

Comments and Suggestions for Authors

In this study, the authors have discussed the catalytic activity of heteroatom doped g-C3N4 particles. They found that B, S, and P doped g-C3N4 can generate more H2 than the undoped g-C3N4 in dehydrogenation of NaBH4 in methanol. In particular, the P doped g-C3N4 showed the best catalytic performance.

  Although the authors discovered interesting results, this paper does not involve fundamental analysis to discuss the catalytic performances. The authors did not chemically specify the catalytic active sites in this dehydrogenation reaction and did not provide quantitative analysis regarding the catalytic active sites in each sample. In this case, it is not possible to do fair-comparison of catalytic performances between different samples. Therefore, I suggest to reject this paper. Please see the more detailed comments below.

-Although we respect the reviewer's point of view, we respectfully disagree with the reviewer’s decision based on our and many other scientists' extensive papers on H2 production from NaBH4 methanolysis utilizing non-metallic catalysts published in many Journals. Also, we would like to get the reviewer's attention on the journal scope as the name of the journal is “C-Journal of Carbon Research” not a catalyst journal.

1)    How much heteroatoms can be incorporated in your synthetic methods? Specify the quantity (e.g. mol %) of heteroatoms in each g-C3N4 catalyst.

-Theoretically, 1:1 doping agents were used for the doping of g-C3N4. However, we could not calculate the specific quantity of heteroatoms in H@g-C3N4 structures for now due to problems with the elemental analysis instrument. However, this will be calculated for our next studies we are planning to use these H@g-C3N4 structures as catalysts in some other reactions e.g., in water splitting. reactions.

2)    Clearly specify and prove the catalytic active sites in this dehydrogenation reaction. Since the undoped-gC3N4 also showed some catalytic performances, the newly added heteroatoms are not just a sole factor to cause this dehydrogenation reaction.

- The following information is provided on pp 11 of the revised manuscript as;

“This work shares the drawbacks encountered in actual applications due to the lack of active sites of bare g-C3N4 [16]. However, by doping with metal-free heteroatoms such as B, S, and P to improve the properties of bare g-C3N4 such as the electrical, functional, and textural characteristics has been proven to overcome these difficulties [41,42]. More active sites may be produced by doping heteroatoms into carbonaceous structures which can improve their catalytic activity by endowing variations in electron density, bond lengths, and atomic sizes depending on the doping agents [43]. Among these heteroatoms, doping P, which has a higher covalent radius than both B and S and has an electronegativity between B and S, can boost the catalytic activity of g-C3N4 more effectively [44].”

3)    Compare the turnover frequency of samples by estimating all of their catalytic active sites. By doing so, you will be able to quantitatively compare their catalytic performances.

- Because of the reason that the catalytic active sites could not be calculated in terms of the specific quantity of heteroatoms in H@g-C3N4 structures, it is not possible to calculate TOF values for the catalyzed reactions.

4)    Prove that the collected gases are pure H2. (by using GC or mass spectrometer)

- In literature, there have been many studies reported using this well-established method by our group and some other research groups about the methanolysis of NaBH4 reactions to produce hydrogen as the reliable measurements for this reaction mechanism [15,16,18, 22, 25, 26, 32]. Therefore, in our opinion, the used method in this manuscript is the most reliable method for the measurement of the produced H2.

5)    Suggest how these heteroatoms can enhance the catalytic performances. In this paper, you just listed numbers proving the reaction results, but you did not provide your insights or proofs for the chemical and physical effects of heteroatoms.

- The following information is provided on pp 11 of the revised manuscript as;

“This work shares the drawbacks encountered in actual applications due to the lack of active sites of bare g-C3N4 [16]. However, by doping with metal-free heteroatoms such as B, S, and P to improve the properties of bare g-C3N4 such as the electrical, functional, and textural characteristics has been proven to overcome these difficulties [41,42]. More active sites may be produced by doping heteroatoms into carbonaceous structures which can improve their catalytic activity by endowing variations in electron density, bond lengths, and atomic sizes depending on the doping agents [43]. Among these heteroatoms, doping P, which has a higher covalent radius than both B and S and has an electronegativity between B and S, can boost the catalytic activity of g-C3N4 more effectively [44].”

Round 2

Reviewer 1 Report

Authors have addressed all the corrections required. I can now recommend their work for publication in C.

Reviewer 3 Report

The paper became more reasonable after revision.